# Association between liberal oxygen therapy and mortality in patients with paraquat poisoning: A multi-center retrospective cohort study

**Xin-Hong Lin[1], Hsiu-Yung Pan[1,2], Fu-Jen Cheng[1,2], Kuo-Chen Huang[1], Chao-Jui Li[1], Chien-Chih Chen[1], Po-Chun Chuang** [1]*

**1** Department of Emergency Medicine, Kaohsiung Chang Gung Memorial Hospital, Chang Gung University College of Medicine, Niaosong Dist., Kaohsiung City, Taiwan (R.O.C.), **2** Chang Gung University College of Medicine, Guishan District, Taoyuan City, Taiwan

\* zhungboqun@gmail.com

## Abstract

Paraquat (N, N'-dimethyl-4, 4'-bipyridinium dichloride, PQ) intoxication is a common cause of lethal poisoning. This study aimed to identify the risk of using liberal oxygen therapy in patients with PQ poisoning. This was a multi-center retrospective cohort study involving four medical institutions in Taiwan. Data were extracted from the Chang Gung Research Database (CGRD) from January 2004 to December 2016. Patients confirmed to have PQ intoxication with a urine PQ concentration $\geq$ 5 ppm were analyzed. Patients who received oxygen therapy before marked hypoxia (SpO2 $\geq$ 90%) were defined as receiving liberal oxygen therapy. The association between mortality and patient demographics, blood paraquat concentration (ppm), and liberal oxygen therapy were analyzed. A total of 416 patients were enrolled. The mortality rate was higher in the liberal oxygen therapy group (87.8% vs. 73.7%, P = 0.007), especially in 28-day mortality (adjusted odds ratio [aOR]: 4.71, 95% confidence interval [CI]: 1.533–14.471) and overall mortality (aOR: 5.97, 95% CI: 1.692–21.049) groups. Mortality in patients with PQ poisoning was also associated with age (aOR: 1.04, 95% CI: 1.015–1.073), blood creatinine level (aOR: 1.49, 95% CI: 1.124–1.978), and blood paraquat concentration (ppm) (aOR, 1.51; 95% CI: 1.298–1.766). Unless the evidence of hypoxia (SpO2 < 90%) is clear, oxygen therapy should be avoided because it is associated with increased mortality.

## Introduction

Paraquat (N, N'-dimethyl-4, 4'-bipyridinium dichloride; PQ) intoxication is a common cause of lethal poisoning in many parts of Asia, Oceania, and the Americas [1, 2]. For example, in Taiwan, 1811 patients were admitted with PQ intoxication from 1997 to 2009, with a mortality rate of 78.6% [3]. Because paraquat is a nonselective, quick-acting, and cheap herbicide, it has been widely used in developing countries [4]. Paraquat is classified as a bipyridyl compound

**Data Availability Statement:** All relevant data are within the manuscript and its Supporting Information files.

**Funding:** The author(s) received no specific funding for this work.

**Competing interests:** The authors have declared that no competing interests exist.

[5]. Its toxicity, which induces nonspecific cellular necrosis, occurs as a result of reactive oxygen species generation [6]. Once paraquat enters the intracellular space, it undergoes a process of alternate reduction and re-oxidation steps known as redox cycling.

Paraquat is oxidized to the paraquat radical upon entry into the cell and is subsequently reduced by enzyme systems such as (Nicotinamide adenine dinucleotide phosphate) NADPH-cytochrome P450 reductase and nitric oxide synthase to form a mono-cation ($PQ^+$) [7–9]. The $PQ^+$ is then rapidly re-oxidized to form the parent paraquat compound in the presence of $O_2$ and generates a superoxide radical (a reactive oxygen species). Reactive oxygen species has the characteristic of cytotoxicity that causes oxidative stress [10–13]. This leads to lipid peroxidation [14, 15], consumption of intracellular NADPH as long as NADPH and oxygen are available [16, 17], mitochondrial damage [18], and even apoptosis [19, 20].

Paraquat causes major organ damage, the most prominent being lung and kidney injuries, since high concentrations of the toxin were found in these organs [16, 21]. Most cases of paraquat ingestion induce poisoning, and the severity of toxicity is related to the dose ingested. The symptoms could be limited and topical if exposure is through dermal contact or through a spray. Lethal complications such as pneumonitis, pulmonary hemorrhage, and acute tubular necrosis could occur [16] if more than 10 mL of the solution (20% wt./vol) is ingested [21]. In previous in vivo studies, supplemental oxygen enhanced the toxicity of paraquat, which resulted in damage to alveolar cells, particularly the type II pneumocytes [22, 23]. In addition, the toxicity seemed to be correlated with the concentration of the oxygen supplied [24, 25]. In clinical practice, emergency physicians do not administer oxygen therapy in patients with acute PQ poisoning unless the patients are hypoxic (usually clinically defined as a pulse oximeter level < 90%) because of the concern that supplemental oxygen might exacerbate the toxicity of paraquat by enhancing the generation of reactive oxygen species [17]. Previous clinical studies have focused on the effects of immunotherapy and hemoperfusion to patients suffering from paraquat poisoning [26, 27]. We conducted a retrospective study to analyze the association between liberal oxygen therapy and the outcomes of PQ poisoning.

## Materials and methods

### Ethics approval

This retrospective study was approved by the Chang Gung medical foundation institutional review board (number 201901558B0). All patient data used in the analyses were anonymized and de-identified.

### Study setting

The data were obtained from the largest health care institution in Taiwan, the Chang Gung Memorial Hospital (CGMH), which receives 10–12% of the National Health Insurance budget according to government statistics. The Chang Gung Research Database (CGRD) was used. This database combines original medical records from four medical institutes (Keelung, Linkou, Chiayi, and Kaohsiung branches) located from northern to southern Taiwan.

### Patients

All patients who experienced paraquat poisoning, visited the emergency department (ED), and had confirmed paraquat intoxication (i.e., urine paraquat concentration $\geq 5$ ppm) from January 2004 to December 2016 were included in the study. Patients who were transferred to other hospitals, discharged against medical advice (DAMA), or exhibited marked hypoxia (SpO2 < 90%) at initial presentation were excluded.

## Measurements

Liberal oxygen therapy was defined as patients receiving oxygen therapy (supplied by a nasal cannula or mask) before marked hypoxia developed (defined as SpO2 ≥ 90%). Conservative oxygen therapy was defined as patients receiving oxygen therapy only if marked hypoxia occurred (defined as SpO2 < 90%). In-hospital mortality and impending death discharge were viewed as mortality [28–30]. The following patient demographics were extracted from the CGRD: age, sex, vital signs, blood creatinine level, urine and blood paraquat concentration (ppm), cyclophosphamide treatment, hemoperfusion, intubation, and signed Do Not Resuscitate (DNR). The paraquat concentration is semi-quantitatively analyzed and the upper limit of this analysis is 50 ppm in urine and 10 ppm in blood.

## Data analysis

For continuous variables with normal distribution: age was summarized as mean ± standard deviation. For continuous variables with non-normal distribution: vital signs, paraquat concentrations, and blood creatinine levels were expressed as medians and first quartiles to third quartiles (Q1-Q3). The distributions of categorical data were presented as numbers and percentages. Student's t-test and the Mann-Whitney U test were used to analyze continuous variables with normal and non-normal distributions, respectively. The chi-square test was used to analyze categorical data. To determine the odds ratios between the variables and mortality, we carried out a multivariate logistic regression. Variables with a $P$-value <0.2 in the univariate analysis between the survival and mortality groups were included in the logistic regression analysis. The effects were estimated in terms of adjusted odds ratios (aORs) with the corresponding 95% confidence intervals (CIs). Results were considered statistically significant for a 2-tailed test if $P < 0.05$. All statistical analyses were performed using SPSS for Windows, version 22.0 (released 2013, IBM Corp., Armonk, NY).

## Results

Fig 1 shows the flowchart of enrollment and the status of patients with PQ poisoning. After excluding patients who were non-critical and DAMA, transferred to other hospitals, or exhibited marked hypoxia at initial presentation, a total of 416 patients were enrolled. The baseline clinical characteristics of patients with PQ poisoning are shown in Table 1. Of the 416 patients who suffered from PQ poisoning, 334 received conservative oxygen therapy and 82 received liberal oxygen therapy. Higher intubation and overall mortality rates were observed in patients who received liberal oxygen therapy ($P = 0.001$ and $P = 0.007$, respectively).

A comparison between the survival group and mortality group (Table 2) showed that the survival group exhibited a younger age (42 ± 14.7 vs. 54 ± 17.1 years, $P < 0.001$) and lower blood paraquat concentration (0.5 [0.1–2] vs. 10 [4.5–10] ppm, $P < 0.001$). The respiratory rate during triage, blood creatinine level, rates of intubation, patients with DNR status, and liberal oxygen therapy administration were also higher in the mortality group.

After analysis with binary logistic regression, the age, blood creatinine, blood paraquat concentration, patients with DNR status, and liberal oxygen therapy were all associated with mortality (Table 3). Fewer patients received cyclophosphamide treatment in the mortality group, but there was no association between cyclophosphamide treatment and mortality (aOR: 1.04, 95% CI: 0.437–2.490).

The adjusted odds ratios and 95% confidence intervals of age, blood paraquat concentration, intubation, and liberal oxygen therapy between different times of mortality are shown in Fig 2. Older age and higher blood paraquat concentrations were associated with higher

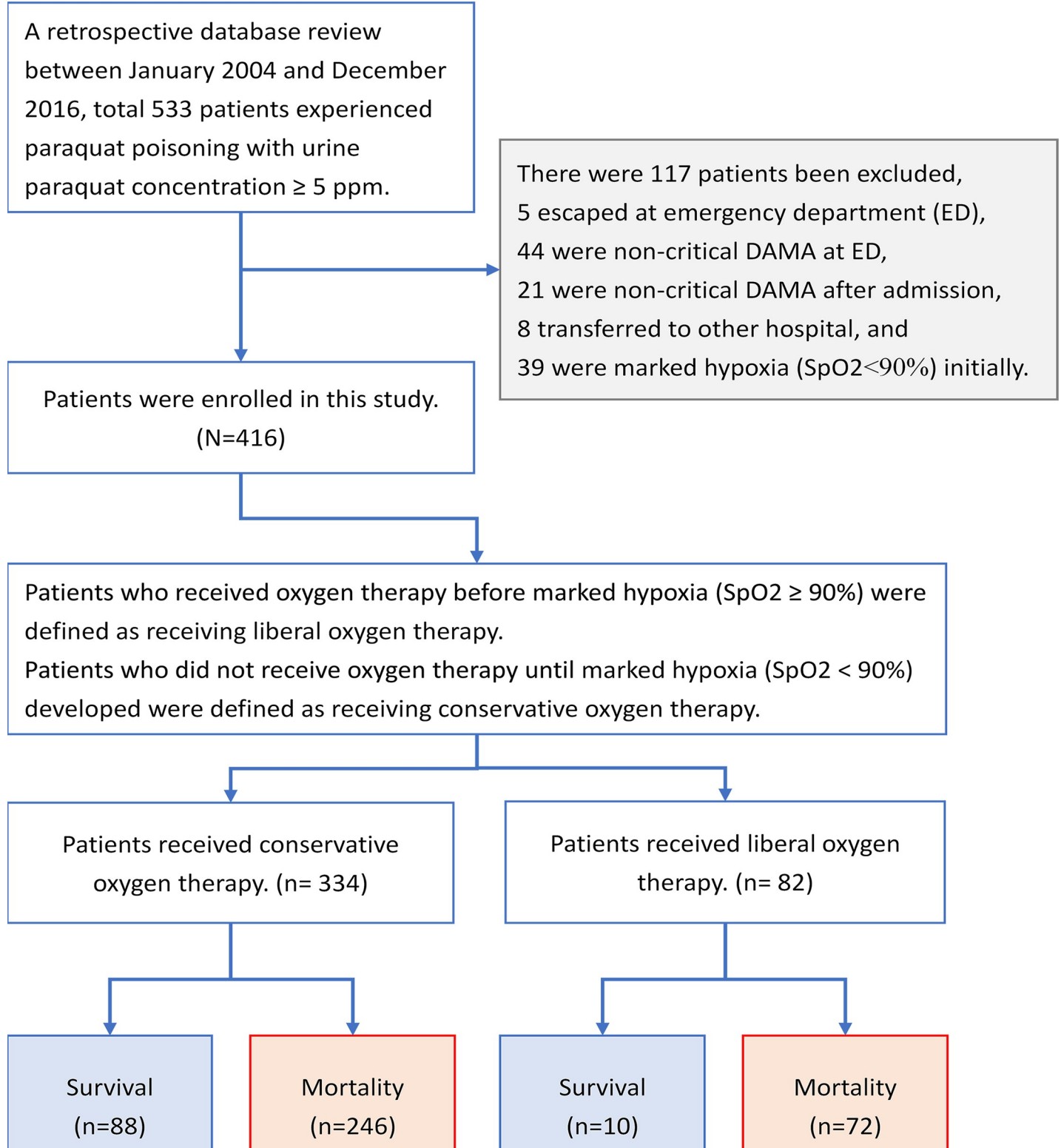

**Fig 1. Flowchart of enrollment and the status of patients upon enrollment.**

**Table 1. Clinical characteristics of patients with paraquat poisoning who received conservative and liberal oxygen therapy (N = 416).**

| | Conservative oxygen therapy n = 334 | Liberal oxygen therapy n = 82 | p-value |
|---|---|---|---|
| Age | 51 ± 17.1 | 55 ± 18.3 | 0.050 |
| Male sex | 244 (73.1) | 51 (62.2) | 0.052 |
| Current smoker | 200 (59.9) | 70 (85.4) | <0.001 |
| Chronic lower respiratory diseases | 2 (0.6) | 4 (4.9) | 0.015 |
| Malignant neoplasms of lung | 0 (0.0) | 1 (1.2) | 0.197 |
| Body temperature during triage (˚C) | 36.1 (35.6–36.7) | 36.0 (35.3–36.6) | 0.181 |
| Heart rate during triage | 92 (79–106.5) | 92.5 (78–107) | 0.948 |
| Respiratory rate during triage | 20 (18–20) | 20 (18–22) | 0.359 |
| Mean arterial pressure during triage | 103.3 (89.7–118.0) | 105.8 (93.3–121.0) | 0.390 |
| SpO2 during triage (%) | 98 (95–100) | 98 (95–100) | 0.351 |
| Glasgow Coma Scale (GCS) | 15 (13–15) | 15 (11–15) | 0.624 |
| Urine paraquat concentration (ppm) | 50 (50–50) | 50 (50–50) | 0.462 |
| Blood paraquat concentration (ppm) | 8.5 (1.5–10) | 8.6 (1.1–10) | 0.950 |
| Blood creatinine level (mg/dL) | 1.8 (1.1–2.8) | 1.7 (1.3–3.6) | 0.252 |
| Cyclophosphamide treatment | 159 (47.6) | 37 (45.1) | 0.687 |
| Hemoperfusion | 236 (70.7) | 58 (70.7) | 0.990 |
| Intubation | 42 (12.6) | 23 (28) | 0.001 |
| Signed Do Not Resuscitate (DNR) | 172 (51.5) | 49 (59.8) | 0.179 |
| Mortality | | | |
| 3-day mortality | 191 (57.2) | 46 (56.1) | 0.858 |
| 7-day mortality | 217 (65) | 59 (72) | 0.231 |
| 28-day mortality | 239 (71.6) | 69 (84.1) | 0.020 |
| Overall mortality | 246 (73.7) | 72 (87.8) | 0.007 |

Data are presented as number (percentage), mean ± SD, or median (Q1-Q3).

Abbreviations: SpO2, peripheral oxygen saturation.

**Table 2. Clinical characteristics of patients between survival and overall mortality patients (N = 416).**

| | Survival patients n = 98 | Mortality patients n = 318 | p-value |
|---|---|---|---|
| Age | 42 ± 14.7 | 54 ± 17.1 | <0.001 |
| Male sex | 64 (65.3) | 231 (72.6) | 0.162 |
| Body temperature during triage (˚C) | 36.5 (36.0–36.9) | 36.0 (35.4–36.5) | <0.001 |
| Heart rate during triage | 91 (79–102) | 92 (79–108) | 0.343 |
| Respiratory rate during triage | 19 (18–20) | 20 (18–22) | 0.001 |
| Mean arterial pressure during triage | 103.3 (92.3–116.0) | 104.2 (89.7–119.0) | 0.966 |
| Glasgow Coma Scale (GCS) | 15 (15–15) | 15 (10–15) | <0.001 |
| Blood paraquat concentration (ppm) | 0.5 (0.1–2) | 10 (4.5–10) | <0.001 |
| Blood creatinine level (mg/dL) | 0.9 (0.7–1.4) | 2 (1.4–3.1) | <0.001 |
| Cyclophosphamide treatment | 60 (61.2) | 136 (42.8) | 0.001 |
| Hemoperfusion | 74 (75.5) | 220 (69.2) | 0.229 |
| Intubation | 4 (4.1) | 61 (19.2) | <0.001 |
| Signed Do Not Resuscitate (DNR) | 14 (14.3) | 207 (65.1) | <0.001 |
| Liberal oxygen therapy | 10 (10.2) | 72 (22.6) | 0.007 |

Data are presented as number (percentage), mean ± SD, or median (Q1-Q3).

**Table 3. Logistic regression analysis of risk factors associated with overall mortality.**

| Variable | aOR | 95% CI of aOR |
|---|---|---|
| Age | 1.04 | 1.015–1.073 |
| Male sex | 2.52 | 0.953–6.649 |
| Current smoker | 0.39 | 0.127–1.168 |
| Chronic lower respiratory diseases | 0.56 | 0.014–22.613 |
| Respiratory rate during triage | 1.20 | 0.991–1.457 |
| Blood paraquat concentration (ppm) | 1.51 | 1.298–1.766 |
| Blood creatinine level (mg/dL) | 1.49 | 1.124–1.978 |
| Cyclophosphamide treatment | 1.04 | 0.437–2.490 |
| Intubation | 4.30 | 1.07–17.303 |
| Signed Do Not Resuscitate (DNR) | 8.50 | 3.347–21.58 |
| Liberal oxygen therapy | 5.97 | 1.692–21.049 |

Abbreviations: aOR, adjusted odds ratio; CI, confidence interval.

The model was adjusted for the following confounders: age, male sex, current smoking, chronic lower respiratory diseases, malignant neoplasms of the lung, body temperature during triage, respiratory rate during triage, Glasgow Coma Scale, blood paraquat concentration (ppm), blood creatinine level (mg/dL), cyclophosphamide treatment, intubation, signed Do Not Resuscitate, and liberal oxygen therapy.

mortality rates. Liberal oxygen therapy was associated with a higher 28-day mortality rate (aOR, 4.71; 95% CI, 1.533–14.471) and a higher overall mortality rate (aOR: 5.97, 95% CI: 1.692–21.049).

The model was adjusted for age, male sex, smoking status (current smoker or not), chronic lower respiratory diseases, malignant neoplasms of the lung, body temperature during triage, respiratory rate during triage, Glasgow Coma Scale, blood paraquat concentration (ppm), blood creatinine level (mg/dL), cyclophosphamide treatment, intubation, signed Do Not Resuscitate, liberal oxygen therapy.

In the subgroup analysis, among the 55 intubated patients with their inspired oxygen fraction (FiO2) recorded after intubation, 26 and 29 used high FiO2 ($\geq$40%) and low FiO2 (<40%) when starting the mechanical ventilator, respectively. The mortality rate was 96.1% and 89.6% in the high and low FiO2 group, respectively ($P = 0.613$). There were 17 patients who initially received high FiO2 in the conservative group (n = 36), and 9 patients initially received high FiO2 in the liberal group (n = 19) (47.2% and 47.4% in the conservative and liberal groups, respectively, $P = 0.992$). There were 10 patients who did not have FiO2 records after intubation due to mortality soon after intubation.

## Discussion

This study involved 416 patients who experienced paraquat intoxication between January 2004 and December 2016 (Fig 1). Mortality rates were 87.8% and 73.7% in the liberal and conservative oxygen therapy groups, respectively (Table 1). Global mortality rates associated with paraquat intoxication have been reported to range from 8% to 78.6% in previous studies [30–32]. The mortality rate was higher in Taiwan (approximately 60%–90%) [33, 34], which could be a result of the accessibility of paraquat, which was not banned in Taiwan until February 2019 [35, 36].

Patients who survived paraquat poisoning were younger (42 ± 14.7 and 54 ± 17.1 years, in the survival and mortality groups, respectively, $P < 0.001$) and exhibited lower blood paraquat concentration (0.5 [0.1–2] vs. 10 [4.5–10] ppm in the survival and mortality groups,

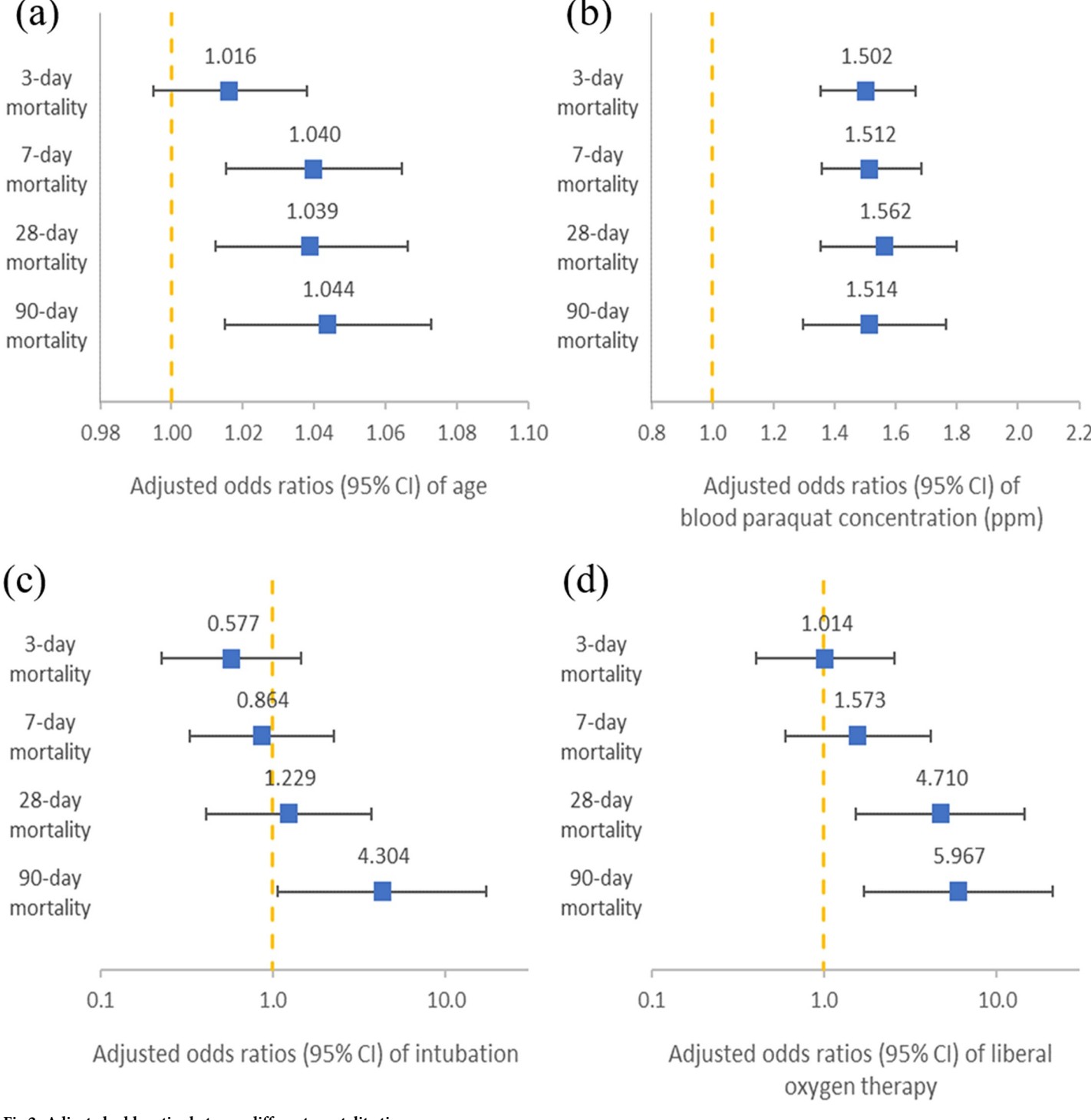

**Fig 2. Adjusted odds ratios between different mortality times.**

respectively, *P* < 0.001) (Table 2). In previous studies, the mortality rate of paraquat intoxication was closely related to age and blood paraquat concentration [30, 37]. The association between mortality and older age, and mortality and higher blood PQ levels were still observed after further analysis of the data with binary logistic regression. Similar to previous studies

[38–41], older ages and higher blood paraquat levels were associated with almost all mortality periods in this study (Fig 2A and 2B).

Mortality was also associated with blood creatinine levels, patients with DNR status, and liberal oxygen therapy (Table 3). Paraquat is primarily eliminated unchanged by the renal system through glomerular filtration and active tubular secretion [42]. It causes acute tubular necrosis, hypoperfusion from hypovolemia/hypotension, and direct glomerular injury following poisoning, which may lead to the development of acute kidney injury [43]. Over 90% of paraquat is excreted in the urine within the first 24 h of poisoning if the renal function is normal [17]. Renal impairment prolongs the elimination of paraquat, which contributes to mortality.

Patients who were intubated due to respiratory failure were associated with a higher overall mortality rate (aOR: 4.30, 95% CI: 1.07–17.303), but were not associated with mortality before 28 days (Fig 2C). In the subgroup analysis among intubated patients, there were 55 patients who had recorded FiO2 levels. The rates of using high FiO2 at the initial phase were not statistically significant between the conservative and liberal groups (47.2% and 47.4%, respectively, $P = 0.992$). The mortality rate was higher in the high FiO2 group (96.1% and 89.6% in the high and low FiO2 groups, respectively, $P = 0.613$), but the difference was not statistically significant. This finding was similar to that of previous studies [3, 44] and might be explained by the different stages of lung injuries caused by paraquat. The initial toxicological effects of paraquat on the lungs are destruction of the alveolar type I and type II epithelial cells, which occur within 1–3 days of poisoning [41, 45–47]. Damage to type I alveolar cells impairs gas exchange between the air space and the capillaries, which compromises lung function from the beginning of paraquat intoxication. The main functions of type II cells are surfactant secretion, active transport of water and ions, and epithelial regeneration. Destruction of type II cells results in increased surface tension within the alveoli, which draws fluid from capillaries to produce edema [48]. The influx of inflammatory cells, mainly neutrophils, macrophages, and eosinophils to the interstitial and alveolar spaces takes place during this destructive phase and is maintained throughout the proliferative phase. Because of this, alveolitis, pulmonary edema, acute pneumonitis, and hemorrhage develop. The proliferative phase, the second phase of paraquat-induced lung toxicity, occurs several days after paraquat ingestion and results in the development of extensive pulmonary fibrosis. The effectiveness of gas exchange is then reduced, which leads to death as a consequence of severe, refractory hypoxia.

As shown in Fig 2D, 28-day mortality and overall mortality (aOR, 4.71; 95% CI, 1.533–14.471; aOR, 5.97; 95% CI, 1.692–21.049, respectively) were associated with liberal oxygen therapy. This may be related to the production of cytotoxic reactive oxygen species. Superoxide radicals are formed by paraquat redox cycling and are susceptible to further reactions by other intracellular processes, leading to the formation of other reactive oxygen species that are also potentially cytotoxic. Paraquat redox cycling continues if nicotinamide adenine dinucleotide phosphate (NADPH) and oxygen are available. Depletion of NADPH prevents the recycling of glutathione and exacerbates toxicity by interfering with other intracellular processes such as energy production and active transport. Intracellular protective mechanisms such as superoxide dismutase and glutathione are also depleted, which further impairs the intracellular clearance of reactive oxygen species. Oxygen supply is believed to amplify the formation of reactive oxygen species. Previous studies also demonstrated that oxygen supply leads to type II pneumocyte injury and impaired pulmonary function [24, 25, 45]. The contribution of liberal oxygen therapy to mortality was more prominent than intubation (aOR = 5.97, $P = 0.005$ and 4.30, $P = 0.04$, respectively, separately shown in Table 3). This might imply that oxygen supply may worsen pulmonary functions and architecture by promoting the process of cytotoxic reactions. Therefore, clinicians should closely monitor oxygen saturation and respiration patterns

in patients with PQ poisoning. Oxygen therapy should be administered with caution and should be reserved for those with hypoxia (SpO2 < 90%).

## Limitations

This study has some limitations. First, we excluded patients who were transferred to other hospitals, escaped or were DAMA. This might have resulted in the higher mortality rates observed in this study. Second, due to the retrospective nature of the study, selection bias cannot be ignored. Patients might have looked much sicker (e.g., shallow breathing or breathing with accessory muscle use), and received "liberal oxygen" even if their oxygen saturation was above 90%. Thus, they ended up having worse conditions. Finally, the limitations of the retrospective design might have introduced some confounding factors that could have altered the values of oxygen saturation. For example, oximeter readings could be influenced by cold extremities or oxygen therapy may be applied by the emergency medical technicians (EMT) outside the hospital.

## Conclusions

Unless the evidence of hypoxia (SpO2 < 90%) is clear, oxygen therapy should be avoided because it is associated with increased mortality.

## Supporting information

**S1 Data.**
(XLSX)

## Acknowledgments

The authors would like to thank the Taiwanese Government for banning paraquat in February 2019 and acknowledge all clinical physicians for their struggle and effort to manage patients with PQ poisoning. The corresponding author Po-Chun Chuang thanks Doctor Ja-Liang Lin who contributed his life to toxicology and medical education in Taiwan.

## Author Contributions

**Conceptualization:** Chao-Jui Li, Po-Chun Chuang.

**Data curation:** Fu-Jen Cheng, Po-Chun Chuang.

**Methodology:** Po-Chun Chuang.

**Supervision:** Hsiu-Yung Pan, Fu-Jen Cheng, Kuo-Chen Huang, Chao-Jui Li, Chien-Chih Chen, Po-Chun Chuang.

**Visualization:** Chien-Chih Chen.

**Writing – original draft:** Xin-Hong Lin.

**Writing – review & editing:** Xin-Hong Lin, Hsiu-Yung Pan.

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
