## [Decision Letter · Decision Letter 0]

22 Sep 2020

PONE-D-20-21510

The association between liberal oxygen therapy and mortality in patients with paraquat poisoning: a multi-center retrospective cohort study

PLOS ONE

Dear Dr. Chuang,

Thank you for submitting your manuscript to PLOS ONE. After careful consideration, we feel that it has merit but does not fully meet PLOS ONE’s publication criteria as it currently stands. Therefore, we invite you to submit a revised version of the manuscript that addresses the points raised during the review process.

I have received the comments of the reviewers on your manuscript. The specific comments of the reviewers are included below. Please provide point by point response in your revised manuscript.

We look forward to receiving your revised manuscript.

Kind regards,

Muhammad Adrish

Academic Editor

PLOS ONE

Journal Requirements:

Reviewers' comments:

Reviewer's Responses to Questions

**Comments to the Author**

1. Is the manuscript technically sound, and do the data support the conclusions?

Reviewer #1: Yes

Reviewer #2: Yes

2. Has the statistical analysis been performed appropriately and rigorously? 

Reviewer #1: Yes

Reviewer #2: Yes

3. Have the authors made all data underlying the findings in their manuscript fully available?

Reviewer #1: Yes

Reviewer #2: Yes

4. Is the manuscript presented in an intelligible fashion and written in standard English?

Reviewer #1: Yes

Reviewer #2: Yes

5. Review Comments to the Author

Reviewer #1: Thanks to the authors for preparing this manuscript on paraquat toxicity and oxygen exposure.

Minor comments:

Abstract – results – can remove the word totally.

28 day – not days.

Introduction : However, there was no clinical evidence …. – there are many publications, back to the 1970s, outlining oxygen toxicity in paraquat exposure. Many of the early papers are in animals. There are multiple publications in humans. Do the authors mean, no previous RCTs looking at oxygen? I do note prior RCTs on therapies (immunosuppression) in paraquat.

Ethics statement - 2. The reason that consent was not obtained (e.g. the data were analyzed

anonymously). ??

Was a waiver of consent provided due to de-identified data being used?

Methods: we used the multivariate analysis, Binary Logistic Regression.

- We carried out a multivariate logistic regression.

Variableswith a p-value <0.2 between the survival and mortality groups were included in the logistic regression analysis.

- Variables with a p-value <0.2 in the univariate analysis ….

Covariates that did not seem to have been considered by the authors, are potentially other patient factors that could make the patient more at risk of hypoxia – such as current smoker, or concurrent background lung disease. Do the authors have these variables available and could put them into the model?

I am interested in the authors choice to compare the survival and mortality groups. Usually you would compare the exposure variable – ie conservative or liberal oxygen. What does table 2 look like done in this way?

Intubation and reduced 3 day mortality – this is probably self fulfilling rather than an actual finding – ie patients sick enough to need intubation, can be maintained early on, but progressive fibrosis gets them later, regardless of intubation. Not sure I would make anything of this result.

Discussion – DNR orders sentence doesn’t seem to be discussing anything and I would remove.

Again, the DNR part of the result, probably reflects a bias whereby those patients not likely to do well (as expected by clinicians) are asked about treatment goals. Interesting, but I wouldn’t stress these results.

Due to the retrospective nature of the study, we can’t ignore selection bias at work here. Patients might have looked sicker, and received oxygen “liberal oxygen”, even if their SaO2 was ok. Thus they end up doing worse.

The intubation effect seen is interesting. You would expect intubated patients to subsequently need more oxygen – as respiratory failure was the reason for intubation. Do the authors have the FiO2 provided to patients who were intubated? It might be interesting to look at the 60 odd intubated patients, and divide them into – low and high FiO2 (><40%) and see if any effect difference seen there.

Reviewer #2: REVIEWER COMMENTS

The authors conducted a retrospective study to determine the relationship between oxygen therapy and the outcomes of paraquat poisoning.

ABSTRACT

1. “Paraquat intoxication-related mortality mainly results from lung injury, which may be associated with oxygen therapy.” This statement in Abstract means what authors sought to investigate is already known.

2. “This study aimed to identify the risk of using oxygen therapy for paraquat poisoning.” The aim is NOT clear.

3. “Totally, 416 patients were enrolled.” Grammatically wrong.

INTRODUCTION

4. “Around 300 000 people die from herbicide or pesticide poisoning each year throughout Asia”. 300,000 should be written well. This statement has no link with previous statement.

5. “The toxicity of paraquat, which induces nonspecific cellular necrosis, results from the generation of reactive oxygen species.” …results from… should be changed to …occurs as a result of…….. The statement should have reference(s).

6. “Reactive oxygen species are potent cytotoxins………….” Uncertain if ROS can be said to be a cytotoxin.

7. “In previous in vivo studies, supplemental oxygen enhanced the toxicity of the paraquat, which resulted in damage to alveolar cells, particularly the type II pneumocytes.” Reference(s) required.

8. “However, there was no clinical evidence that oxygen therapy would worsen the prognosis in patients with paraquat poisoning.” If this statement can be clarified. Pratt et al (1980), Bateman & Leach (1998), and others, have shown that oxygen therapy enhances the toxic effects of paraquat poisoning.

METHODS

9. …………..or exhibited marked hypoxia (SpO2 < 90%) at initial presentation were excluded. Explain this statement; as some patients in the study had SpO2 <90%.

10. ……………..were extracted from the CGRD database. I believe D is acronym is database.

11. Under measurements: last line; ………hemoperfusion, intubation, and signed DNR. Write DNR in full.

12. The Student’s t test and Mann-Whitney u test were used to analyze the data. Which data? And what was the use of these inferential statistics?

RESULTS

13. Table 1 has a number of issues;

• Title needs to be improved (include sample size)

• Body temperature needs to be at 1 d.p

• Urine paraquat concentration cannot be 50 + 0.0 for the 2 groups. The p value is erroneous

• Table needs to show SpO2 for the two groups

• How can one differentiate between mean (SD) and median (quartile deviation)

14. Table 2 has some few isssues;

• Body temperature needs to be at 1 d.p

• Glasgow Coma Scale (GCS): p < 0.001? How is that possible?

15. Why didn’t the adjusted odds ratio include DNR? Indeed; the highest OR was with DNR: could that be determinant for mortality vis-à-vis oxygen therapy?

16. Legend for Figure 1 needs to be improved (more details; retrospective data, period, etc)

17. Figure 1: Explain in box using SpO2 “conservative oxygen therapy” and “liberal oxygen therapy”

DISCUSSION

18. “Mortality rates are 87.8% and 73.7% in liberal and conservative oxygen therapy groups, respectively (Table 1).” Mortality rates were……

6. PLOS authors have the option to publish the peer review history of their article (what does this mean?). If published, this will include your full peer review and any attached files.

Reviewer #1: No

Reviewer #2: No

---

## [Author Response · Author response to Decision Letter 0]

13 Nov 2020

Dear Academic Editor: Muhammad Adrish

Thank you for giving us the opportunity to submit the revision of “The association between liberal oxygen therapy and mortality in patients with paraquat poisoning: a multi-center retrospective cohort study” for your consideration for publishing in the PLOS ONE.

Thanks to the reviewers for their careful review. The reviewers’ comments have been carefully read and responded to as following. 

We hope the finding and conclusion described in our manuscript may benefit patients who got paraquat intoxication worldwide, and we also hope the precious readers of PLOS ONE will find this study interesting. We are hopeful that this manuscript will meet the standards of your prestigious journal and be favorably reviewed.

We look forward to hearing from you again soon. 

Sincerely yours,

Corresponding author: 

Dr. Po-Chun Chuang, MD

Department of Emergency Medicine 

Kaohsiung Chang Gung Memorial Hospital

No. 123, Dapi Rd., Niaosong Dist. Kaohsiung 833, Taiwan, R.O.C.

E-mail: zhungboqun@gmail.com. 

 

Dear reviewer #1: 

1. Abstract – results – can remove the word totally.

Response: Thank you for your suggestion. The sentence has been rewritten to make it clearer. In order to make the article much smoother, we sent the manuscript to the language editing service.

2. 28 day – not days.

Response: Thank you for your kind reminder. We have corrected this grammatical mistake.

3. Introduction: However, there was no clinical evidence …. – there are many publications, back to the 1970s, outlining oxygen toxicity in paraquat exposure. Many of the early papers are in animals. There are multiple publications in humans. Do the authors mean, no previous RCTs looking at oxygen? I do note prior RCTs on therapies (immunosuppression) in paraquat.

Response: Thank you for recommendation. No previous RCTs looking at liberal and conservative oxygen therapy was founded. Probably because oxygen therapy is considered harmful. Therefore, we try to constructive a retrospective study to analyze the association. The sentence was re-written as below.

“Previous clinical studies have focused on the effects of immunotherapy and hemoperfusion to patients suffering from paraquat poisoning [26, 27]. However, the association between liberal oxygen therapy and prognosis in patients with PQ poisoning is still unclear. Therefore, we conducted a retrospective study to analyze the association between liberal oxygen therapy and the outcomes of PQ poisoning.”

4. Ethics statement - 2. The reason that consent was not obtained (e.g. the data were analyzed anonymously)?

Was a waiver of consent provided due to de-identified data being used?

Response: Thank you for your friendly reminder. Yes, the data were analyzed anonymously. The Chang Gung Research Database (CGRD), the multi‐institutional electronic medical records collection in Taiwan. Because all data converted from the original electronic medical records were anonymized, the study protocol was approved, and informed consents were exempted by the Institutional Review Board of Chang Gung Memorial Hospital (IRB No: 201901558B0). 

5. Methods: we used the multivariate analysis, Binary Logistic Regression.

- We carried out a multivariate logistic regression.

Variables with a p-value <0.2 between the survival and mortality groups were included in the logistic regression analysis.

- Variables with a p-value <0.2 in the univariate analysis ….

Response: Thank you for your recommendation. The sentence has been rewritten to make it clearer.

6. Covariates that did not seem to have been considered by the authors, are potentially other patient factors that could make the patient more at risk of hypoxia – such as current smoker, or concurrent background lung disease. Do the authors have these variables available and could put them into the model?

Response: Thank you for your kind reminder. Current smoker, chronic lower respiratory diseases, and malignant neoplasms of lung were added as variables and put into the logistic regression model.

7. I am interested in the authors choice to compare the survival and mortality groups. Usually you would compare the exposure variable – ie conservative or liberal oxygen. What does table 2 look like done in this way?

Response: Thank you for your kind reminder. We compared the exposure variable in table 1. We compared the survival and mortality groups in table 2 because we want to see if there are variables that significantly affect mortality rate. 

8. Intubation and reduced 3-day mortality – this is probably self fulfilling rather than an actual finding – ie patients sick enough to need intubation, can be maintained early on, but progressive fibrosis gets them later, regardless of intubation. Not sure I would make anything of this result.

Response: Thank you for this recommendation. After adjusting more variables (current smoker, chronic lower respiratory diseases, and malignant neoplasms of lung), intubation is not associated with 3-day mortality.

9. Discussion – DNR orders sentence doesn’t seem to be discussing anything and I would remove.

Again, the DNR part of the result, probably reflects a bias whereby those patients not likely to do well (as expected by clinicians) are asked about treatment goals. Interesting, but I wouldn’t stress these results.

Response: Thank you for your recommendation. The paragraph of DNR in discussion was removed.

10. Due to the retrospective nature of the study, we can’t ignore selection bias at work here. Patients might have looked sicker, and received oxygen “liberal oxygen”, even if their SaO2 was ok. Thus, they end up doing worse.

Response: Thank you for this important and professional recommendation. This point is added in limitation to make the retrospective study clearer.

Below sentences was added in limitation.

“Second, due to the retrospective nature of the study, selection bias cannot be ignored. Patients might have looked much sicker (e.g., shallow breathing or breathing with accessory muscle use), and received “liberal oxygen” even if their oxygen saturation was above 90%. Thus, they ended up having worse conditions.”

 

11. The intubation effect seen is interesting. You would expect intubated patients to subsequently need more oxygen – as respiratory failure was the reason for intubation. Do the authors have the FiO2 provided to patients who were intubated? It might be interesting to look at the 60 odd intubated patients and divide them into – low and high FiO2 (><40%) and see if any effect difference seen there.

Response: Thanks for this great recommendation. The subgroup analysis for intubated patients was done. Among 65 intubated patients, 55 patients had been recorded FiO2 data and 10 patients had not been recorded FiO2 data due to rapid expired after intubation. The finding was added in result and discussion. 

 

Dear reviewer #2: 

ABSTRACT

1. “Paraquat intoxication-related mortality mainly results from lung injury, which may be associated with oxygen therapy.” This statement in Abstract means what authors sought to investigate is already known.

Response: Thanks for recommendation. This sentence is deleted.

2. “This study aimed to identify the risk of using oxygen therapy for paraquat poisoning.” The aim is NOT clear.

Response: Thank you for your kind reminder. Although the pathophysiology of paraquat has been studied, there are only a few studies researching on human being. The aim of our study is to find if the liberal oxygen therapy in patients who were got paraquat intoxication is consistent with the previous animal studies. The sentence has been rewritten as below to make it clearer.

“This study aimed to identify the risk of using liberal oxygen therapy in patients with PQ poisoning.”

3. “Totally, 416 patients were enrolled.” Grammatically wrong.

Response: Thank you for your recommendation. The sentence has been rewritten to make it clearer.

INTRODUCTION

4. “Around 300 000 people die from herbicide or pesticide poisoning each year throughout Asia”. 300,000 should be written well. This statement has no link with previous statement.

Response: Thank you for your kind reminder. The sentence has been rewritten to make it clearer. 

5. “The toxicity of paraquat, which induces nonspecific cellular necrosis, results from the generation of reactive oxygen species.” …results from… should be changed to …occurs as a result of…….. The statement should have reference(s).

Response: Thank you for your recommendation. The sentence has been rewritten and reference has also been cited.

6. “Reactive oxygen species are potent cytotoxins………….” Uncertain if ROS can be said to be a cytotoxin.

Response: Thank you for your kind reminder. The sentence has been rewritten as “Reactive oxygen species has the characteristic of cytotoxicity that causes oxidative stress.”. 

7. “In previous in vivo studies, supplemental oxygen enhanced the toxicity of the paraquat, which resulted in damage to alveolar cells, particularly the type II pneumocytes.” Reference(s) required.

Response: Thank you for your recommendation. The reference has also been cited.

8. “However, there was no clinical evidence that oxygen therapy would worsen the prognosis in patients with paraquat poisoning.” If this statement can be clarified. Pratt et al (1980), Bateman & Leach (1998), and others, have shown that oxygen therapy enhances the toxic effects of paraquat poisoning.

Response: Thank you for your recommendation. The sentence has been rewritten as below.

“However, the association between liberal oxygen therapy and prognosis in patients with PQ poisoning is still unclear.” 

METHODS

9. …………..or exhibited marked hypoxia (SpO2 < 90%) at initial presentation were excluded. Explain this statement; as some patients in the study had SpO2 <90%.

Response: Thank you for your kind reminder. 

In our study, we excluded the patients who got the severe hypoxia (SpO2<90%) at “initial presentation” (such as in the triage). In our study, patients with severe hypoxia (SpO2 <90%) during hospitalization were initially saturated with greater than 90%.

10. ……………..were extracted from the CGRD database. I believe D is acronym is database.

Response: Thank you for your friendly reminder. The repeated word “database” has been deleted.

11. Under measurements: last line; ………hemoperfusion, intubation, and signed DNR. Write DNR in full.

Response: Thank you for your kind reminder. The abbreviation has been rewritten as “Do Not Resuscitation” 

12. The Student’s t test and Mann-Whitney u test were used to analyze the data. Which data? And what was the use of these inferential statistics?

Response: Thanks for recommendation. To make it clear, the sentences was rewritten as below.

“For continuous variables with normal distribution: age was summarized as mean ± standard deviation. For continuous variables with non-normal distribution: vital signs, paraquat concentrations, and blood creatinine levels were expressed as medians and first quartiles to third quartiles (Q1-Q3). The distributions of categorical data were presented as numbers and percentages. Student’s t-test and the Mann-Whitney U test were used to analyze continuous variables with normal and non-normal distributions, respectively. The chi-square test was used to analyze categorical data.”

RESULTS

13. Table 1 has a number of issues;

• Title needs to be improved (include sample size)

Response: Thank you for your kind reminder. The sample size is added in title.

• Body temperature needs to be at 1 d.p

Response: Thank you for your kind reminder. The table has been corrected.

• Urine paraquat concentration cannot be 50 + 0.0 for the 2 groups. The p value is erroneous

Response: Thank you for your kind reminder. In Chang Gung Memorial Hospital system, the urine paraquat concentration is semi-quantitative analysis and the upper limit of this analysis is 50(ppm). If the concentration is above 50 ppm, the accurate value cannot be quantified and shown as 50 ppm in result. To make it clear, below sentence will be added in method paragraph.

“The paraquat concentration is semi-quantitatively analyzed and the upper limit of this analysis is 50 ppm in urine and 10 ppm in blood.”

• Table needs to show SpO2 for the two groups

Response: OK thanks for recommendation.

• How can one differentiate between mean (SD) and median (quartile deviation)

Response: Thanks for this important recommendation. The non-normal distribution continuous variables are expressed in median (Q1-Q3) to make it clearer. 

14. Table 2 has some few isssues;

• Body temperature needs to be at 1 d.p

Response: Thank you for your kind reminder. The table has been corrected.

• Glasgow Coma Scale (GCS): p < 0.001? How is that possible?

Response: Thank you for your friendly reminder. To make it clear, the non- continuous variables are expressed in median (Q1-Q3). GCS will be expressed as 15 (15-15) in survival group and 15 (10-15) in mortality group. I think it is the reason that the p-value is <0.001. Thanks again for the wonderful recommendation.

15. Why didn’t the adjusted odds ratio include DNR? Indeed; the highest OR was with DNR: could that be determinant for mortality vis-à-vis oxygen therapy?

Response: Thank you for your recommendation. 

Thank you for your recommendation. The adjusted odds ratios are included DNR and other variables. Therefore, the table 3 was rewritten.

16. Legend for Figure 1 needs to be improved (more details; retrospective data, period, etc)

Response: Thanks for recommendation. Figure 1 has been improved.

17. Figure 1: Explain in box using SpO2 “conservative oxygen therapy” and “liberal oxygen therapy”

Response: Thanks for recommendation. Figure 1 has been improved.

DISCUSSION

18. “Mortality rates are 87.8% and 73.7% in liberal and conservative oxygen therapy groups, respectively (Table 1).” Mortality rates were……

Response: Thank you for your kind reminder. The sentence has been rewritten.

---

## [Decision Letter · Decision Letter 1]

25 Nov 2020

PONE-D-20-21510R1

Association between liberal oxygen therapy and mortality in patients with paraquat poisoning: a multi-center retrospective cohort study

PLOS ONE

Dear Dr. Chuang,

Thank you for submitting your manuscript to PLOS ONE. After careful consideration, we feel that it has merit but does not fully meet PLOS ONE’s publication criteria as it currently stands. Therefore, we invite you to submit a revised version of the manuscript that addresses the points raised during the review process.

ACADEMIC EDITOR: Please see attached comments by the reviewers. Kindly provide point by point response in your revised manuscript.

We look forward to receiving your revised manuscript.

Kind regards,

Muhammad Adrish

Academic Editor

PLOS ONE

Reviewers' comments:

Reviewer's Responses to Questions

**Comments to the Author**

1. If the authors have adequately addressed your comments raised in a previous round of review and you feel that this manuscript is now acceptable for publication, you may indicate that here to bypass the “Comments to the Author” section, enter your conflict of interest statement in the “Confidential to Editor” section, and submit your "Accept" recommendation.

Reviewer #1: (No Response)

Reviewer #2: All comments have been addressed

2. Is the manuscript technically sound, and do the data support the conclusions?

Reviewer #1: Partly

Reviewer #2: Yes

3. Has the statistical analysis been performed appropriately and rigorously? 

Reviewer #1: Yes

Reviewer #2: Yes

4. Have the authors made all data underlying the findings in their manuscript fully available?

Reviewer #1: Yes

Reviewer #2: Yes

5. Is the manuscript presented in an intelligible fashion and written in standard English?

Reviewer #1: Yes

Reviewer #2: Yes

6. Review Comments to the Author

Reviewer #1: Thanks to the authors for their revisions.

I note you still present the fact that no association between liberal oxygen and outcome in paraquat poisoning has been presented before. I do not think this is true and both reviewers provided multiple references to show this.

Line 183 - signing DNR consents, - this should be something like "patients with DNR status"

Suggest changing throughout

Line 228 - pulmonary function (not s) by causing cytotoxic reactions.

Line 231 - Conclusion: Oxygen therapy should be administered with caution and should be reserved for those with hypoxia (SpO2 < 90 %)."

Isn't that what happened???

Oxygen requirement is a signal for poor outcome.

Reviewer #2: (No Response)

7. PLOS authors have the option to publish the peer review history of their article (what does this mean?). If published, this will include your full peer review and any attached files.

Reviewer #1: No

Reviewer #2: No

---

## [Author Response · Author response to Decision Letter 1]

5 Dec 2020

Response to the Editor and the Reviewer

Dear Academic Editor: Muhammad Adrish

Thank you for giving us the opportunity to submit the revision of “The association between liberal oxygen therapy and mortality in patients with paraquat poisoning: a multi-center retrospective cohort study” for your consideration for publishing in the PLOS ONE.

Thanks to the reviewers for their careful review. The reviewers’ comments have been carefully read and responded as following. 

We hope the finding and conclusion described in our manuscript may benefit patients who got paraquat intoxication worldwide, and we also hope the precious readers of PLOS ONE will find this study interesting. We are hopeful that this manuscript will meet the standards of your prestigious journal and be favorably reviewed.

We look forward to hearing from you again soon. 

Sincerely yours,

Corresponding author: 

Dr. Po-Chun Chuang, MD

Department of Emergency Medicine 

Kaohsiung Chang Gung Memorial Hospital

No. 123, Dapi Rd., Niaosong Dist. Kaohsiung 833, Taiwan, R.O.C.

E-mail: zhungboqun@gmail.com. 

 

Dear reviewer #1: 

1. I note you still present the fact that no association between liberal oxygen and outcome in paraquat poisoning has been presented before. I do not think this is true and both reviewers provided multiple references to show this.

Response: Thank you for your kind reminder. The sentence has been removed.

2. Line 183 - signing DNR consents, - this should be something like "patients with DNR status" Suggest changing throughout

Response: Thank you for your kind reminder. There are thee “signing DNR consents” in our article and we have changed them to “patients with DNR status”. 

3. Line 228 - pulmonary function (not s) by causing cytotoxic reactions.

Response: Thank you for your friendly reminder. The sentence has been rewritten to make it clearer.

4. Line 231 - Conclusion: Oxygen therapy should be administered with caution and should be reserved for those with hypoxia (SpO2 < 90 %)." Isn't that what happened??? Oxygen requirement is a signal for poor outcome.

Response: Thank you for your kind reminder. It is true that oxygen requirement is a signal for poor outcome. In addition, this study revealed the association between the liberal oxygen therapy and mortality in patients with paraquat poisoning. However, this study could only reveal the correlation, not causation. The causation between liberal oxygen therapy and mortality need to be proved by further prospective studies and randomized control trials. However, the ethical and legal issues restrict the conduction of prospective studies since oxygen therapy are harmful for paraquat poisoning patients. To make it clear, the conclusion was rewritten as “Unless the evidence of hypoxia (SpO2 < 90%) is clear, oxygen therapy should be avoided because it is associated with increased mortality.”

Reviewer #2: (No Response)

---

## [Decision Letter · Decision Letter 2]

29 Dec 2020

Association between liberal oxygen therapy and mortality in patients with paraquat poisoning: a multi-center retrospective cohort study

PONE-D-20-21510R2

Dear Dr. Chuang,

We’re pleased to inform you that your manuscript has been judged scientifically suitable for publication and will be formally accepted for publication once it meets all outstanding technical requirements.

Kind regards,

Muhammad Adrish

Academic Editor

PLOS ONE

Additional Editor Comments (optional):

You have addressed all the queries raised by the reviewers.

Reviewers' comments:

Reviewer's Responses to Questions

**Comments to the Author**

1. If the authors have adequately addressed your comments raised in a previous round of review and you feel that this manuscript is now acceptable for publication, you may indicate that here to bypass the “Comments to the Author” section, enter your conflict of interest statement in the “Confidential to Editor” section, and submit your "Accept" recommendation.

Reviewer #1: All comments have been addressed

2. Is the manuscript technically sound, and do the data support the conclusions?

Reviewer #1: Yes

3. Has the statistical analysis been performed appropriately and rigorously? 

Reviewer #1: Yes

4. Have the authors made all data underlying the findings in their manuscript fully available?

Reviewer #1: Yes

5. Is the manuscript presented in an intelligible fashion and written in standard English?

Reviewer #1: Yes

6. Review Comments to the Author

Reviewer #1: (No Response)

7. PLOS authors have the option to publish the peer review history of their article (what does this mean?). If published, this will include your full peer review and any attached files.

Reviewer #1: No

---

## [Editor Report · Acceptance letter]

7 Jan 2021

PONE-D-20-21510R2 

Association between liberal oxygen therapy and mortality in patients with paraquat poisoning: a multi-center retrospective cohort study 

Dear Dr. Chuang:

I'm pleased to inform you that your manuscript has been deemed suitable for publication in PLOS ONE. Congratulations! Your manuscript is now with our production department. 

Kind regards, 

on behalf of

Dr. Muhammad Adrish 

Academic Editor

PLOS ONE